# Epigenetic Clock Explains White Matter Hyperintensity Burden Irrespective of Chronological Age

**DOI:** 10.3390/biology12010033

**Published:** 2022-12-24

**Authors:** Joan Jiménez-Balado, Eva Giralt-Steinhauer, Isabel Fernández-Pérez, Lucía Rey, Elisa Cuadrado-Godia, Ángel Ois, Ana Rodríguez-Campello, Carolina Soriano-Tárraga, Uxue Lazcano, Adrià Macias-Gómez, Antoni Suárez-Pérez, Anna Revert, Isabel Estragués, Brigitte Beltrán-Mármol, Santiago Medrano-Martorell, Jaume Capellades, Jaume Roquer, Jordi Jiménez-Conde

**Affiliations:** 1Neurovascular Research Group, Department of Neurology, Institut Hospital del Mar d’Investigacions Mèdiques (IMIM), 08003 Barcelona, Spain; 2Medicine Department, DCEXS-Universitat Pompeu Fabra (UPF), 08002 Barcelona, Spain; 3Department of Psychiatry, NeuroGenomics and Informatics, Washington University School of Medicine, St. Louis, MO 63110, USA; 4Unidad de Investigación AP-OSIs, 20014 Donostia, Spain; 5Radiology Department, Neuroradiology Unit, Hospital del Mar, 08003 Barcelona, Spain

**Keywords:** white matter hyperintensities, cerebral small vessel disease, epigenetics, DNA methylation, biological age, epigenetic clock

## Abstract

**Simple Summary:**

Biological age (B-age), or the degree of aging of an individual, can differ from chronological age. B-age is affected by epigenetics, and we calculate it based on the degree of methylation of multiple specific regions of human DNA. For previous research, we know that patients with ischemic stroke are biologically older than healthy individuals without stroke. On the other hand, white matter hyperintensities (WMH) observed in brain magnetic resonance images are an unspecific sign that has been associated with brain aging and also with the increased risk of stroke or dementia. It is unknown whether epigenetic biological age is associated with this sign of brain aging. In this manuscript, we interrogated the association between B-age and WMH volume and found that patients with high WMH burden are biologically older. Moreover, we found that 42.7% of the effects of chronological age on WMH can be explained by B-age, suggesting a role of epigenetics in WMH pathophysiology. Our study also generates a potential number of questions that might be addressed in further articles, such as whether this relationship depends on WMH location.

**Abstract:**

In this manuscript we studied the relationship between WMH and biological age (B-age) in patients with acute stroke. We included in this study 247 patients with acute stroke recruited at Hospital del Mar having both epigenetic (DNA methylation) and magnetic resonance imaging data. WMH were measured using a semi-automated method. B-age was calculated using two widely used methods: the Hannum and Horvath formulas. We used multiple linear regression models to interrogate the role of B-age on WMH volume after adjusting for chronological age (C-age) and other covariables. Average C-age of the sample was 68.4 (±11.8) and we observed a relatively high median WMH volume (median = 8.8 cm^3^, Q1–Q3 = 4.05–18.8). After adjusting for potential confounders, we observed a significant effect of B-age_Hannum_ on WMH volume (β_Hannum_ = 0.023, *p*-value = 0.029) independently of C-age, which remained significant (β_C-age_ = 0.021, *p*-value = 0.036). Finally, we performed a mediation analysis, which allowed us to discover that 42.7% of the effect of C-age on WMH is mediated by B-age_Hannum_. On the other hand, B-age_Hoarvath_ showed no significant associations with WMH after being adjusted for C-age. In conclusion, we show for the first time that biological age, measured through DNA methylation, contributes substantially to explain WMH volumetric burden irrespective of chronological age.

## 1. Introduction

White matter hyperintensities (WMH) are considered the main hallmark of cerebral small vessel disease (cSVD), which encompasses all those vascular risk factors that affect the small vessels of the brain [1]. WMH are a common radiological marker observed in older individuals, having an estimated prevalence at any degree ranging from 39 to 96% [2]. Exposition to risk factors, especially hypertension, conditions the progression of WMH, which are estimated to increase in a rate between 0.2 and 1.2 cm^3^/year and are even higher in patients with ischemic stroke or with vascular risk factors [3]. Even being subclinical, this progression is associated with incident stroke, cognitive impairment and gait disturbances [4].

Although we know the consequences of WMH, their pathophysiology is not fully understood. Clinical studies have shed light on which clinical variables relate to increased WMH burden, being age and high blood pressure; factors that have been consistently associated with WMH [5]. Furthermore, several studies have explored whether blood biomarkers that were altered in stroke patients also correlated with the WMH burden [6,7]. However, most groups have focused on proteomic studies [7], while epigenetics of WMH have received less attention. Epigenetics (the regulation of gene expression without altering the DNA sequence) might provide insight on which pathological mechanisms trigger WMH accumulation. 

DNA methylation (DNAm) is an epigenetic mechanism regulating high-order DNA structure and gene expression by the addition of a methyl group to the five-carbon position of cytosine in a cytosine–phosphate–guanine (CpG) context [8]. It has been widely studied in aging research, given that DNAm varies across the lifespan. This variation has been used to create estimators of chronological age (C-age) through multiple CpGs across the genome [9,10]. We usually refer to this calculation as biological age (B-age), and the mismatch between C-age and B-age can be considered as a surrogate marker of age acceleration. We previously reported that patients with stroke are biologically older than patients without [11]. Moreover, we also proved that B-age is a better predictor than C-age of stroke outcome, mortality or recurrence [12,13,14]. However, less is known on the role of B-age in subclinical cerebrovascular disease and WMH burden. 

In this study we aimed to interrogate the association between WMH and B-age in patients with acute stroke admitted at the Hospital del Mar. Additionally, we tested whether B-age is a better predictor of WMH burden than C-age. 

## 2. Methods

### 2.1. Participants and Setting

This study is nested in the BasicMar cohort, which is an ongoing observational and prospective register of patients diagnosed with acute stroke at the Hospital del Mar (Barcelona) that includes genetic and neuroimaging databanks [15]. 

For this study we used cross-sectional data of patients recruited from 2009 to 2013 having the following inclusion criteria: (1) diagnosis of ischemic stroke (not infrequent causes), (2) availability of genetic and DNAm data [15,16], (3) MRI examination including at least fluid attenuated inversion recovery (FLAIR) and diffusion-weighted images (DWI) and (4) not having other neurological or genetic diseases or toxic habits that could confound the evaluation of WMH (e.g., multiple sclerosis). A total of 247 individuals met these criteria. 

This study was approved by the institutional review board at the Hospital del Mar and all patients, or legal representatives, provided a written informed consent prior to the study enrollment. This study was in agreement with the Helsinki Declaration. 

### 2.2. DNA Methylation Array

DNA was isolated from whole peripheral blood collected in 10 mL EDTA tubes using the Chemagic Magnetic Separation Module I system (Chemagen). DNA extractions were obtained at the same time after the acute event and stored at minus 20 °C. DNA concentrations and quality were checked with PicoGreen assay and agarose gels, respectively.

Genome-wide DNA methylation data were obtained in three technical runs corresponding to three independent studies. The first two batches (N = 428 and N = 232) were analyzed using the Human Methylation 450 K Beadchip with 485,577 CpG sites (CpGs) and the third (N = 379) with the Infinium Methylation EPIC Beadchip consisting of 865,918 probes (Illumina Netherlands, Eindhoven, The Nethterlands). We followed the manufacturer’s protocol in both cases. The arrays corresponding to these three batches were scanned with the Illumina HiScan SQ scanner at Progenika Biopharma in Bizkaia, Spain. 

Data were then processed using standard pipelines [17]. Briefly, intensity data files were loaded using the R-library *Minfi.* Then, we calculated β values, which range from 0 (completely unmethylated) to 1 (completely methylated) [18]. Regarding sample QC, we removed those samples showing detection rates < 98% or those presenting a sex mismatch. For CpGs, we excluded those probes having detection *p*-values > 0.05 in at least 1% of samples. Additionally, those CpGs having a count lower than 3 in at least 5% of samples were filtered as well. We subsequently normalized the filtered β values using the beta-mixture quantile normalization method [19] and corrected the batch effect with the *sva* library [20]. Finally, we calculated the white blood cell counts from DNAm data using the Houseman algorithm [21]. 

### 2.3. Biological Age Estimation 

B-age was calculated using two different well-described methods: the Hannum and Horvath formulas [9,10]. Hannum’s method uses the methylation of 71 CpGs to predict, while Horvath’s method uses 353. We lacked some of these probes either due to failed quality controls or due to CpGs that were not included in the EPIC array (we had available 58/71 and 326/353 for Hannum and Horvath’s formulas, respectively). However, this had a minimum effect in the B-age estimation, as shown in Figure 1, where we observe a correlation between B-age estimations and C-age close to 0.8. Estimations of B-age using only those CpGs included in the EPIC array have demonstrated to be equivalent and accurate for the Hannum and Horvath estimators [22].

### 2.4. Neuroimaging

#### 2.4.1. Acquisition

MRI scans were acquired on 1.5-T or 3-T scanners (GE medical systems and Philips Achieva 3.0T X-Series MRI System, respectively) as part of routine clinical practice. For this study, we used clinical axial 2-D FLAIR-weighted images (fluid-attenuated inversion recovery). In most cases, images were acquired on the 3-T scanner with TE/TR of 125/11,000 ms, flip angle of 90 degrees, in-plane matrix size of 512 × 512 and consisted of 25 slices with a slice thickness of 5 mm and 0.4 mm gap. Those images obtained on the 1.5-T had a TE/TR of 156/10,000, flip angle of 90 degrees, in-plane matrix size of 512 × 512 and consisted of 20 slices with a slice thickness of 5 mm and 1.5 mm gap.

#### 2.4.2. WMH Volume Quantification

WMH volume (WMHv) analysis was performed on axial FLAIR sequences using a MRIcro software, according to previously validated methods [23]. FLAIR and the DWI sequences were aligned to exclude acute and chronic infarcts. Using operator-mediated quality assurances, overlapping regions of interest (ROIs) corresponding to WMH produced the final maps for WMH volume calculation. To correct WMH volume for head size, we used the sagittal midline cross-sectional intracranial area (ICA) as a surrogate measure of the intracranial volume. WMH was normalized multiplying the measured WMH by the ratio of the individual ICA to mean ICA of the whole cohort. Supratentorial WMH volume was given as log-transformed. All supratentorial white matter and deep grey matter lesions were included, with the exception of WMH corresponding to infarcts. To further avoid confusion, we measured only WMH from the hemisphere unaffected by stroke and doubled this value to calculate total WMHv. All readers have previously shown a high interrater agreement for the determination of WMHv (free marginal k > 0.90). All MRI measurements were performed by readers blinded to clinical data.

### 2.5. Clinical Variables

Trained neurologists collected the demographic information, as well as vascular risk factors and clinical data, using standardized forms during the hospitalization. For this study we used: age, sex, hypertension, diabetes mellitus, hyperlipidemia, body mass index, current smoking habit, alcohol consumption, coronary disease, atrial fibrillation and stroke etiology. Stroke subtypes were categorized according to the TOAST system into large-artery atherosclerosis, cardioembolism, small-vessel occlusion (lacunar), stroke of other determined etiology and stroke of undetermined etiology. However, no patient with the subtype “other etiologies” was included in this study, as described in Section 2.1. The complete protocol has been published elsewhere [24].

### 2.6. Statistics

Descriptive data were reported as means (±SD), medians (Q_1_–Q_3_) or frequencies (%), according to the type and distribution of each variable. As WMHv presented a skewed distribution, we used the log_e_ transformation (log-WMHv). 

First, we conducted a set of bivariate analyses aimed to find which variables were statistically associated with log-WMHv. Pearson’s and Spearman correlation coefficients, *t*- or anova tests were used appropriately to this aim, according to each case. 

We then constructed general linear models aimed to test the relationship between log-WMHv and B-age, calculated both with Hannum’s and Horvath’s formulas, independently of C-age and other covariables. To that aim, we constructed three separate models for B-age_Hannum_ and B-age_Horvath_. Model 1 was adjusted for C-age and hypertension. Model 2 was additionally adjusted for those variables significantly associated with WMH in the bivariate analyses or which have been described to have an effect on WMH in the previous literature [25]. Thus, we adjusted for sex, diabetes, hypercholesterolemia, smoking habit, alcohol consumption, stroke etiology and body mass index. Model 3 was Model 2 plus adjustment for principal components summarizing white cell count estimates, which might influence the methylation signal [21]. We used principal components, because white cell counts are highly correlated and might be a source of concerning multicollinearity and bias. Principal component analysis was conducted using the correlation matrix of CD8T-, CD4T-, NK-, monocyte- and granulocyte-cell estimates (we excluded the estimate showing the lower cell proportion in the sample: B-cell estimates). To extract the best number of variables we used the parallel analysis algorithm, which compares the *scree* plots from actual and simulated data (95% percentile of simulated eigen values) [26]. Finally, variables from Model 3 were selected using a forward stepwise algorithm based on the Aikake index criterion (AIC). In this last model, B-age_Hannum_ and B-age_Horvath_ were introduced together in the scope model in order to find which of these variables had a stronger effect on WMH. All statistical assumptions were checked and met. Concerning multicollinearity was checked by calculating variance inflation factors (VIF) and tolerance (1/VIF) measurements. Presence of influential cases was checked by calculating Cook’s distance.

Finally, we conducted a mediation analysis to understand the inter-relationships between C-age, B-age and WMH. Briefly, a mediation analysis tested the hypothesis of whether the effect of an independent variable (C-age) on the dependent variable (WMHv) is partially or totally explained by a third variable (B-age, DNAm). Here, we used the causal steps approach, where there must be a direct effect between the independent variable (C-age) and the outcome (WMHv), an effect between the independent variable and the mediator (B-age) as well as an adjusted effect between the mediator and the outcome [27]. The confidence interval of the mediated effect was obtained after 1000 bootstrapped simulations and it corresponded to the 2.5 and 97.5 percentiles of this bootstrapped distribution. 

## 3. Results

### 3.1. Principal Characteristics of the Cohort

Summary statistics from our sample are presented in Table 1. As expected, we observed a relatively high average WMHv in the sample (median = 8.8 cm^3^, Q_1_–Q_3_ = 4.05–18.8). 

In the bivariate analyses, we found a significant effect of age, hypertension, diabetes, smoking habit and alcohol consumption on WMHv, such that older patients with hypertension and diabetes showed increased WMHv (Appendix A). 

In Figure 1, we show the correlation between C-age and B-age estimations (both Hannum and Horvath methods). Both B-age_Hannum_ and B-age_Horvath_ showed strong correlations with C-age (r = 0.790 and r = 0.761, respectively), suggesting that both formulas had a good fit in the sample regarding the prediction of C-age. As shown in Figure 2, C-age and both B-age calculations showed significant correlations with WMHv (r = 0.36–0.38, *p*-value < 0.0001 in all cases). 

### 3.2. Effect of B-Age on WMH

In Table 2, we show the results obtained from multivariate analyses interrogating the association between B-age and log-WMHv. Regarding the Hannum method, in Model 1 (only adjusted by hypertension and C-age), we observed that B-age showed a borderline effect (*p*-value = 0.058), while C-age was significantly associated with log-WMHv (β = 0.020, 95% CI = 0.003 to 0.038, *p*-value = 0.021). However, when we further adjusted for other potential confounders that are known to affect WMH, we observed that B-age_Hannum_ had a significant effect on log-WMHv (β = 0.022, 95% CI = 0.002 to 0.043, *p*-value = 0.031) and C-age remained significant as well (β = 0.021, 95% CI = 0.002 to 0.040, *p*-value = 0.033). The addition of principal components representing the white cellular count estimates did not change these results (Model 3, Table 2). On the other hand, B-age_Horvath_ presented no significant associations (*p*-values > 0.05 in all models). We observed no concerning multicollinearity in any model (VIF < 4 and 1/VIF < 0.2 in all cases).

We continued simplifying Model 3 using a forward stepwise method (Table 3). This algorithm selected the following variables: B-age_Hannum_, C-age, sex at birth, TOAST (stroke etiology), hypertension, diabetes, smoking habit, alcohol consumption and body mass index. It excluded: B-age_Horvath_, dyslipidemia and white cellular count estimates. In this model, B-age_Hannum_ showed an independent association with log-WMHv (β = 0.023, 95% CI = 0.002 to 0.043, *p*-value = 0.029), such that, for each year increase in B-age_Hannum,_ we observed a 2.3% increase in WMHv. Again, we observed no concerning collinearity. 

As both B-age and C-age remained in the final model, we explored the amount of the effect of C-age on WMH that can be explained by DNAm. We used a mediation analysis to explore this hypothesis and we observed a significant indirect effect of B-age on WMHv via B-age_Hannum_. Specifically, we observed that a 42.7% of the C-age effect on WMH can be explained by B-age_Hannum_ (partial mediation), suggesting an important role of DNAm on WMH burden (Figure 3). 

## 4. Discussion

In this manuscript we explored the association between B-age and WMH, finding a significant effect of B-age on WMH, which is independent of C-age. When we explored which variables entered in a forward stepwise regression model to predict WMH, we observed that both B-age_Hannum_ and C-age entered the final model, being independent explanatory factors. Finally, as both variables contributed to the prediction of WMH, we hypothesized that a part of the effect of C-age on WMHv would be explained by B-age_Hannum_. Hence, we conducted a mediation analysis and found that a 42.7% of the effect of C-age can be explained by DNAm. 

The main contribution of our study is to show a relationship between WMHv and epigenetic aging independently of C-age. To our better knowledge, only one group has previously explored the role of biological aging on WMH burden, but they studied a cohort of African-American individuals at high cardiovascular risk, evaluating WMH by means of a qualitative assessment [28]. Our study aligns with the results obtained by Raina and collaborators (2017) and extends their associations to our cohort of Caucasian patients with acute ischemic stroke, in which we evaluated WMH as a volume using a semi-automated method. On the other hand, while B-age_Hannum_ was independently associated with WMHv, B-age_Horvath_ was not. This might be explained by that fact that B-age_Hannum_ is calculated using whole blood—as we do in our study—as opposed to the B-age_Horvath_ formula, which was constructed on a range of different tissues and cell types [10]. 

B-age explains 42.7% of the contribution of C-age to WMH burden but, at the same time, contributes to WMH prediction independently of C-age. This means that B-age might be capturing the effect of other factors that lead to an epigenetic signature similar to that observed in the elderly, as we proposed in a previous article reporting that ischemic stroke individuals are biologically older than healthy controls [11]. Supporting this hypothesis, B-age has been previously associated with systolic blood pressure and arterial stiffness [29], as measured by pulse pressure, which are considered to be the main blood pressure components involved in sporadic cSVD [5,30]. Similarly, several consequences of WMH have been consistently associated with B-age. For instance, a study conducted in a sample of healthy adults found that B-age_Hannum_ was associated with decline in attention in men [31], which is one of the cognitive domains affected in patients having an extensive WMH burden [32]. Likewise, patients with incidental cardiovascular disease showed an increased age acceleration compared with patients without [33]. 

These studies, along with the results provided in our manuscript, suggest the idea that several exposures —vascular risk factors, lifestyle, comorbid conditions—might influence the epigenome of individuals having an increased burden of WMH. Hence, it would be potentially interesting to conduct large epigenome-wide association studies (EWAS) aimed to find hypo/hypermethylated regions of the human genome that could shed light on new mechanisms involved in the pathophysiology of WMH and eventually represent new therapeutic targets. To date, only one study has conducted an EWAS using the WMHv as dependent variable, although it found no significant results, likely due to a reduced sample size [34]. Similarly, other groups measuring WMH using qualitative approaches also failed in reporting significant CpGs [28]. Future studies should try to increase the sample size and consider other factors that might confound the relationship between DNAm and WMH burden. For instance, the location of WMH within the brain might correspond to different pathophysiological mechanisms, as reported recently [4,35,36], and thus present different risk factors and epigenetic signatures. To this aim, we may need to combine EWAS with new neuroimaging analytical approaches that improve the ability to capture the true nature of WMH [37,38].

A major strength of this study is that it evaluates the volumetric WMH burden through a semiautomated method that captures this phenomenon in a more accurate way than qualitative discrete scales. In addition, the independent analyses of B-age and C-age, together with the mediation analysis, can help in understanding their different contributions. Some limitations of the study should be considered. We measured methylation levels in peripheral blood-cell DNA and, for some CpGs, the methylation is tissue-specific [39]. Therefore, we could have lost signals by not choosing distinct tissues where epigenetic age may have a higher repercussion on WMH. However, methylation patterning of whole blood has been described as a good approach for the methylation of a specific location [40]. We also make clear that in this cross-sectional study we cannot establish causality in its associations.

## 5. Conclusions

In conclusion, we show for the first time that B-age, measured through DNAm, contributes substantially to explain WMH volumetric burden irrespective of C-age. In a mediation analysis, we observed that B-age explains 42.7% of the effect of C-age on WMH volume. Our study provides further insight into the pathophysiology of WMH, but it also generates a potential number of hypotheses and questions that might be approached in future studies. For instance, whether the relationship between B-age and WMH is dependent on brain location or radial distribution, especially considering that this distribution is related to specific consequences [38]. Furthermore, these neuroimaging techniques might be combined with the full genome-wide study of DNAm. Moreover, disentangling how biological age captures other vascular and external factors’ contribution to WMH also merits further research on this matter, as it might provide orientation in which factors could be modified to halt the progression of WMH.

## Figures and Tables

**Figure 1 biology-12-00033-f001:**
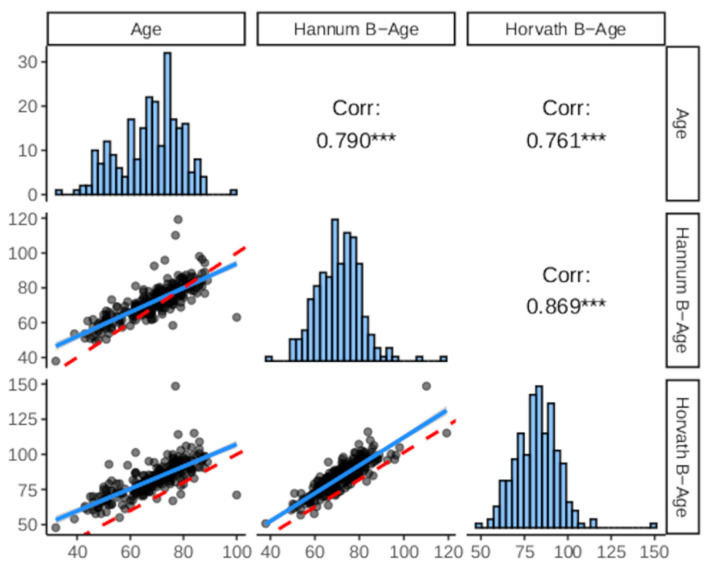
Correlation between biological age estimations and chronological age. Lower and upper panels represent the correlation between biological age estimations (Hannum and Horvath formulas) and chronological age.The red-dashed lines in the lower panels represent the “perfect” concordance between variables, while the blue straight lines correspond to the true association obtained from a linear fit. Diagonal panels show the histograms from each variable. *** *p*-value < 0.001.

**Figure 2 biology-12-00033-f002:**
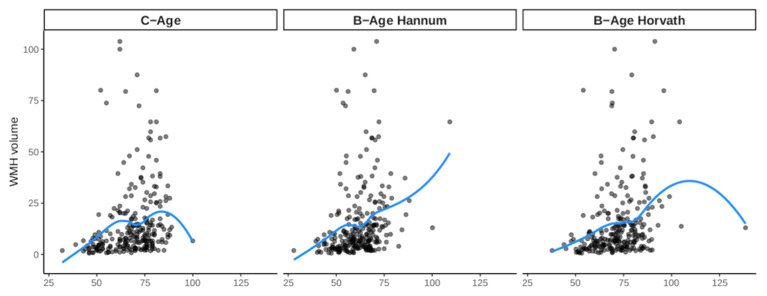
Bivariate association between chronological and biological age estimations and WMH volume. The blue line represents the “loess” fit in each case.

**Figure 3 biology-12-00033-f003:**
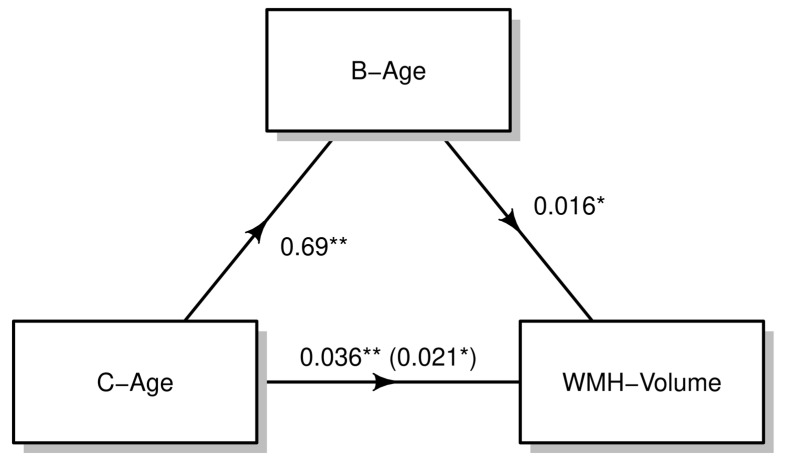
Causal mediation analysis. After bootstrap resampling, C-age showed a significant effect on B-age (β = 0.7), which was subsequently related to WMH-volume (β = 0.016). Mediation analysis revealed a significant partial mediation on the effect of C-age on WMH via B-age (β-coefficient reduction from 0.036 to 0.021). These effects are adjusted for those variables that entered in the forward stepwise regression model (see Table 3). * *p*-value < 0.05; ** *p*-value < 0.01.

**Table 1 biology-12-00033-t001:** Data are presented as frequencies and percentages or mean and standard deviations.

Principal Characteristics of the Sample (N = 247)	
Variable	Mean (SD)/N(%)
Age, years	68.4 (11.8)
Sex, male	155 (62.8%)
Smoking habit, yes	85 (34.4%)
Alcohol consumption	67 (27.1%)
Hypertension	192 (77.7%)
Diabetes	103 (41.7%)
Dyslipidemia	149 (60.3%)
Body mass index	
Normal, <25	79 (34.1%)
Overweight, 25 to 30	98 (42.2%)
Obesity, ≥30	55 (23.7%)
Atrial fibrillation	65 (26.3%)
Previous myocardial infarction	22 (8.94%)
TOAST, stroke subtype	
Atherothrombotic	62 (25.1%)
Lacunar	83 (33.6%)
Cardioembolic	59 (23.9%)
Undetermined	43 (17.4%)

**Table 2 biology-12-00033-t002:** All models have been constructed entering log-WMHv as the dependent variable and biological age estimations and chronological age as independent variables. Model 1: adjusted for C-age and hypertension. Model 2: Model 1 + sex, diabetes, hypercholesterolemia, smoking habit, alcohol consumption, stroke etiology (TOAST) and body mass index. Model 3: Model 2 + white cell count estimates (principal components). Values represent β coefficients, 95% confidence intervals (CI), *p*-values and valid observations in each model.

	Effect of Biological Age in WMH							
	Hannum Models				Horvath Models			
	Biological Age		Chronological Age		Biological Age		Chronological Age	
	β (95% CI)	*p*-Value	β (95% CI)	*p*-Value	β (95% CI)	*p*-Value	β (95% CI)	*p*-Value
**Model 1**	0.019 (−0.001; 0.038)	0.058	0.020 (0.003; 0.038)	0.021	0.014 (−0.002; 0.030)	0.078	0.023 (0.006; 0.039)	0.007
**Model 2**	0.022 (0.002; 0.043)	0.031	0.021 (0.002; 0.040)	0.033	0.018 (0,000; 0.035)	0.054	0.024 (0.005; 0.042)	0.012
**Model 3**	0.022 (0.001; 0.042)	0.039	0.021 (0.002; 0.040)	0.033	0.017 (−0.001; 0.035)	0.060	0.023 (0.005; 0.042)	0.013

**Table 3 biology-12-00033-t003:** This model was achieved after applying a forward stepwise variable selection on Model 3 from Table 2. Of note, B-age_Hannum_ rather than B-age_Horvath_ was included in the final model. Values represent β coefficients, 95% confidence intervals (CI) and *p*-values.

	Effect of B-Age and Other Risk Factors on WMH	
	β (95% CI)	*p*-Value
B-age (Hannum), year	0.023 (0.002; 0.043)	0.029
C-age, year	0.021 (0.001; 0.040)	0.036
Sex, male	0.286 (−0.003; 0.575)	0.053
TOAST, stroke subtype		
Atherothrombotic	Ref.	Ref.
Cardioembolic	−0.135 (−0.513; 0.243)	0.481
Lacunar	0.339 (0.001; 0.677)	0.050
Undetermined	0.118 (−0.270; 0.506)	0.550
Hypertension	0.375 (0.059; 0.692)	0.020
Diabetes	0.295 (0.034; 0.557)	0.027
Smoking Habit	0.341 (−0.026; 0.709)	0.069
Alcohol Consumption	−0.444 (−0.799; −0.089)	0.014
Body Mass Index	−0.026 (−0.057; 0.004)	0.095

## Data Availability

Data will be shared upon petition of qualified researchers.

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
