# Peer review of "Epigenetic Clock Explains White Matter Hyperintensity Burden Irrespective of Chronological Age"

_biology, 2022, doi:10.3390/biology12010033_

Round 1

Reviewer 1 Report

This study examined the association of biological age, estimated using the Hannum and Horvath DNA methylation age assessment methods, with volume of white matter hyperintensities (WMHs) in 247 individuals with acute ischemic stroke. The authors reported a significant effect of biological age, estimated using the Hannum method, but not the Horvath method, on WMHs that is independent of chronic age after adjusting for confounders. The paper is well written and easy to follow, and as the authors stated, provides a novel association between epigenetic aging and WMH volume in Caucasian patients with acute ischemic stroke. I have some minor comments listed below.

1.      Caption of figure 2: WMH volume in figure is not log-transformed but is stated as log-transformed. Is the individual aged ~100 an outlier because there are no other individuals aged close to 100? I am wondering whether the Horvath estimation of biological age will improve after removing this individual.

2.      Tables 1 & 3: what is the “undetermined” variable? Please also explain “TOAST” for readability.

3.      Would the authors explain why biological age is older than chronological aging in younger patients, but not in older patients as it is shown in figure 2?

Author Response

We first want to acknowledge the time and efforts that Reviewer-1 spent in our manuscript. His/her comments have improved the quality of our work.

Please, find reviewer’s comments in bold text. Our answers will be displayed in regular font and changes in the main text in red font.

This study examined the association of biological age, estimated using the Hannum and Horvath DNA methylation age assessment methods, with volume of white matter hyperintensities (WMHs) in 247 individuals with acute ischemic stroke. The authors reported a significant effect of biological age, estimated using the Hannum method, but not the Horvath method, on WMHs that is independent of chronic age after adjusting for confounders. The paper is well written and easy to follow, and as the authors stated, provides a novel association between epigenetic aging and WMH volume in Caucasian patients with acute ischemic stroke. I have some minor comments listed below.

We are happy to read that Reviewer-1 found our manuscript of interest.

1. Caption of figure 2: WMH volume in figure is not log-transformed but is stated as log-transformed.

The reviewer is right. This is a typographic error, thank you for pointing it out. We corrected it. In the visual representation we preferred to plot the raw WMH volume because we consider that it is easier to interpret for readers. Moreover, this plot just represents the bivariate association between WMH and biological age (B-age). In the multivariate analyses, we indeed used the log-transformed WMH volume to meet the statistical assumptions required for a linear regression model.

Is the individual aged ~100 an outlier because there are no other individuals aged close to 100? I am wondering whether the Horvath estimation of biological age will improve after removing this individual.

This is an interesting point and a fair question. In the final model (table 3), as well as in previous models (table 2), we checked the presence of influential cases calculating the Cook’s distance from each observation. For Horvath’s models included in table 2 we see the following distribution of influential cases (model 1 to 3, from left to right):

Cook’s distance (Y axis) suggests concerning collinearity for values higher than 1[1]⁠. As observed, we saw no concerning influence in our models. However, some cases (IDs 2444, 3135, 3259, 4655) are slightly deviated from the sample average. Therefore, we’ve decided to explore the WMH volume, Horvath and Hannum’s B-Age estimations as well as chronological age from these cases:

ID

Age

Horvath’s B-Age

Hannum’s B-Age

WMH, cc

2444

77

138.5

100.1

13

3135

62

91.3

71.0

103.8

3259

52

54.0

50.1

80

4655

67

63.6

50.1

32

As observed in the table, the case 2444 corresponds to a subjects showing an average WMH volume, but a high age acceleration. This is observed both for Hannum’s and Horvath’s formulas, but the mismatch is higher for Horvath’s clock. This is the case that the reviewer noticed in the plot. However, this case only shows a certain degree of influence in the first model (the left plot from the previous image).

Although according to Cook’s distance we have no influential cases in our dataset, we followed the Reviewer’s recommendation and repeated all models displayed in table 2. These results are showed in the table on the next page. We only observed a change in Model-1 results, where we see that B-AgeHannum and B-AgeHorvath became significant after excluding this particular case. However, model 2 and 3 show the same results, suggesting that when we consider those clinical factors associated with WMH, only B-AgeHannum has a significant correlation with WMH volume irrespective of chronological age, as exposed in the original article. We consider that this might be explained by the better fit of Hannum’s clock in our sample, in which epigenetic data was obtained in whole blood samples as Hannum and colleagues did in their study[2]⁠. On the other hand, Horvath’s formula was obtained via a multi-tissue cohort[3]⁠.

Although we came to the same conclusions and results after excluding this subject, and even knowing that this subject is going against the association show in our results, we consider that it is more correct to keep results in the current form. It is indeed a patient with a high age acceleration, corroborated by both formulas, but who shows a lower WMH volume. But this is completely possible. Considering that epigenetic clocks are affected by a wide range of factors, including environmental, lifestyle and socioeconomic factors[4], different circumstances may explain discrepancies between B-Age and expected WMH in a particular individual (such as the exposure to factors that may affect intensely epigenetics in a very quick way, but without sufficient time to affect the neuroimaging).

However, we think that it is important to add a sentence in the statistics section explaining that we checked influential cases in our model:

All statistical assumptions were checked and met. Concerning multicollinearity was checked by calculating variance inflation factors (VIF) and tolerance (1/VIF) measurements. Presence of influential cases was checked by calculating Cook’s distance. (page: 5, paragraph: 4)

Effect of biological age in WMH after excluding outlier case

Hannum Models

Horvath Models

Biological age

Chronological age

Biological age

Chronological age

β (95% CI)

p-value

β (95% CI)

p-value

β (95% CI)

p-value

β (95% CI)

p-value

Model 1

0.0210(0,000;0.042)

0.045

0.019(0.001;0.037)

0.036

0.018(0;0.036)

0.047

0.020(0.002;0.037)

0.027

Model 2

0.022(0.002;0.043)

0.031

0.021(0.002;0.04)

0.033

0.018(0;0.035)

0.054

0.024(0.005;0.042)

0.012

Model 3

0.022(0.001;0.042)

0.039

0.021(0.002;0.040)

0.033

0.017(-0.001;0.035)

0.060

0.023(0.005;0.042)

0.013

All models have been constructed entering log-WMHv as the dependent variable and biological age estimations and chronological age as independent variables.

Model-1: adjusted for C-age and hypertension.

Model-2: Model-1+sex, diabetes, hypercholesterolemia, smoking habit, alcohol consumption, stroke etiology (TOAST) and body-mass index.

Model-3: Model-2+white cell count estimates (principal components).

Values represent β coefficients, 95% confidence intervals (CI), p-values and valid observations in each model.

Tables 1 & 3: what is the “undetermined” variable? Please also explain “TOAST” for readability.

We agree, we forgot to add a brief explanation of what TOAST is in the methods section. TOAST is a classification of stroke subtypes according to etiology. It classifies stroke cases in one of the following categories: 1) large-artery atherosclerosis, 2) cardioembolism, 3) small-vessel occlusion (lacunar), 4) stroke of other determined etiology, and 5) stroke of undetermined etiology. An undetermined (or cryptogenic) stroke means that it was not possible to know the cause or etiology of a stroke, and they tend to represent up to ¼ of stroke cases. We added the following description in the methods section:

Stroke subtypes were categorized according to TOAST system into: large-artery atherosclerosis, cardioembolism, small-vessel occlusion (lacunar), stroke of other determined etiology, and stroke of undetermined etiology. (page 5, paragraph 1).

And we also changed table 1 & 3 labels to: TOAST, stroke subtype.

Would the authors explain why biological age is older than chronological aging in younger patients, but not in older patients as it is shown in figure 2?

This is an interesting question and common effect observed in epigenetic studies interrogating the effects of B-age, and it is mainly explained through the “survival bias”. In our study, additionally, another reasons may contribute to intensify this effect:

1) This is a sample of stroke patients. And, as we reported in a previous manuscript, patients with stroke are biologically older than matched controls (subjects without stroke and same chronological age)[5]⁠. At the same time, stroke incidence increases with increasing age.

2) B-age proved to be associated with mortality risk[6]⁠. Therefore, in our sample we probably have a survivor-bias in older patients.

Therefore, some younger patients might exhibit a higher B-age as compared to older individuals because they are stroke patients, and it is uncommon to suffer stroke at the middle age, so they probably had a poor lifestyle or vascular health. On the other hand, older patients, even if they had a stroke, are probably subjects who have a relatively good health and, thus, there is a survivor bias in this subjects because they arrived at the older age.

1. Olson DL. 2020 Predictive data mining models. Second edition. Singapore: Springer.

2. Hannum G et al. 2013 Genome-wide Methylation Profiles Reveal Quantitative Views of Human Aging Rates. Mol. Cell 49, 359–367. (doi:10.1016/j.molcel.2012.10.016)

3. Horvath S, Raj K. 2018 DNA methylation-based biomarkers and the epigenetic clock theory of ageing. Nat. Rev. Genet. 19, 371–384. (doi:10.1038/s41576-018-0004-3)

4. Drew L. 2022 Turning back time with epigenetic clocks. Nature 601, S20–S22. (doi:10.1038/d41586-022-00077-8)

5. Soriano-Tárraga C et al. 2016 Ischemic stroke patients are biologically older than their chronological age. Aging (Albany. NY). 8, 2655–2666. (doi:10.18632/aging.101028)

6. Fransquet PD, Wrigglesworth J, Woods RL, Ernst ME, Ryan J. 2019 The epigenetic clock as a predictor of disease and mortality risk: A systematic review and meta-analysis. Clin. Epigenetics 11, 1–17. (doi:10.1186/s13148-019-0656-7)

Reviewer 2 Report

This is a well-structured research article. The main question addressed by this research is the fact that the extent of white matter lesions of the brain depends on DNA methylation. I think that this is an original and interesting topic for the readers of this journal.

This paper adds to this scientific area as it is the first article showing that biological age, measured through DNA methylation, affects the white matter hyperintensity, irrespective of chronological age.

The introduction gives the background of this study as it describes the white matter hyperintensity and its correlation with SSVD, as well as the DNA methylation as an epigenetic mechanism.

“Materials and Methods” section is descriptive enough and well-structured too. It refers to the patients studied, ethical aspects, the DNA methylation array, the method of biological age’s calculation, neuroimaging methods and statistical analyses used in this study.

The results are very interesting and, to my opinion, well presented.

The discussion is well written, summarizing and discussing the main findings of the study and describing also its main limitations, which is important.

Conclusions although consistent with the evidence presented, perhaps could be written in a separate section and in a more detailed manner, presenting and analyzing some specific targets for future studies, that have been mentioned in the discussion.

References are relative to the subject and sufficient in number.

English language and style are generally fine. Minor issues need to be addressed before publication (e.g. in line 170 the word “test” should be replaced by “tests”).

Author Response

We first want to acknowledge the time and efforts that Reviewer-1 spent in our manuscript.

Please, find reviewer’s comments in bold text. Our answers will be displayed in regular font and changes in the main text in red font.

This is a well-structured research article. The main question addressed by this research is the fact that the extent of white matter lesions of the brain depends on DNA methylation. I think that this is an original and interesting topic for the readers of this journal.

This paper adds to this scientific area as it is the first article showing that biological age, measured through DNA methylation, affects the white matter hyperintensity, irrespective of chronological age.

The introduction gives the background of this study as it describes the white matter hyperintensity and its correlation with SSVD, as well as the DNA methylation as an epigenetic mechanism.

Materials and Methods” section is descriptive enough and well-structured too. It refers to the patients studied, ethical aspects, the DNA methylation array, the method of biological age’s calculation, neuroimaging methods and statistical analyses used in this study.

The results are very interesting and, to my opinion, well presented.

The discussion is well written, summarizing and discussing the main findings of the study and describing also its main limitations, which is important.

References are relative to the subject and sufficient in number.

Conclusions although consistent with the evidence presented, perhaps could be written in a separate section and in a more detailed manner, presenting and analyzing some specific targets for future studies, that have been mentioned in the discussion.

We thank the reviewer for the comments. We agree with what it was suggested, so we incorporated in the revised version, expanded conclusions in a separate section:

In conclusion, we show for the first time that B-age, measured through DNAm, contributes substantially to explain WMH volumetric burden, irrespective of C-age. In a mediation analysis we observed that B-age explains 42.7% of the effect of C-age on WMH volume. Our study provides further insight about the pathophysiology of WMH, but it also generates a potential number of hypotheses and questions that might de approached in future studies. For instance, whether the relationship between B-age and clustering WMH is dependent on brain location or radial distribution, especially considering that this distribution is related to specific consequences[38]. Besides, these neuroimaging techniques might be combined with the full genome-wide study of DNAm. Moreover, disentangle how biological age captures other vascular and external factors contribution to WMH also merits further research on this matter, as it might provide orientation in which factors could be modified to halt the progression of WMH. (page: 11, paragraph: 5).

English language and style are generally fine. Minor issues need to be addressed before publication (e.g. in line 170 the word “test” should be replaced by “tests”).

Thank you for noticing this issue. We corrected this and other typographic/grammar mistakes and we hope that the current version of our manuscript is ready for publication.

Reviewer 3 Report

Manuscript deals with the association between WMH and B-age in patients with acute stroke. The authors previously reported that patients with stroke are biologically older than patients without and also proved that B-age is a better predictor than C-age of stroke outcome, mortality or recurrence. The authors tested whether B-age is a better predictor of WMH burden than C-age. Results of the study showed for the first time that biological age, measured through DNA methylation, contributes substantially to explain WMH volumetric burden, irrespective of  chronological age.

The manuscript is obviously of great interest, although authors should be kinky asked to comment on some major issues.

The most important issue that should be thoroughly revised is MRI protocol description. It is recommended to provide more complete sequences description including slice thickness, flip angle, dist factor, examination time etc., e.g.

Also, I would recommend to describe in details WMH volume analysis with MRIcro software in 2.4 Neuroimaging.

Author Response

We first want to acknowledge the time and efforts that Reviewer-3 spent in our manuscript. His/her comments have improved the quality of our work, highlighting several aspects that needed to be explained in detail.

Please, find reviewer’s comments in bold text. Our answers will be displayed in regular font and changes in the main text in red font.

Manuscript deals with the association between WMH and B-age in patients with acute stroke. The authors previously reported that patients with stroke are biologically older than patients without and also proved that B-age is a better predictor than C-age of stroke outcome, mortality or recurrence. The authors tested whether B-age is a better predictor of WMH burden than C-age. Results of the study showed for the first time that biological age, measured through DNA methylation, contributes substantially to explain WMH volumetric burden, irrespective of chronological age.

The manuscript is obviously of great interest, although authors should be kinky asked to comment on some major issues.

We are glad to know that Reviewer-3 found our manuscript of interest and we are sure that it clearly improved after incorporating his/her suggested modifications.

The most important issue that should be thoroughly revised is MRI protocol description. It is recommended to provide more complete sequences description including slice thickness, flip angle, dist factor, examination time etc., e.g.

Following the recommendation we expanded information about the neuroimaging protocol. We added the following section:

2.4. Neuroimaging

2.4.1. Acquisition

MRI scans were acquired on 1.5-T or 3-T scanners (GE medical systems and Philips Achieva 3.0T X -Series MRI System, respectively) as part of routine clinical practice. For this study we used clinical axial 2-D FLAIR-weighted images (Fluid-Attenuated Inversion Recovery). In most cases, images were acquired on the 3-T scanner with TE/TR of 125/11,000 ms, flip angle of 90 degrees, in-plane matrix size of 512×512 and consisted of 25 slices with a slice thickness of 5mm and 0.4mm gap. Those images obtained on the 1.5-T had a TE/TR of 156/10,000, flip angle of 90 degrees, in-plane matrix size of 512×512 and consisted of 20 slices with a slice thickness of 5mm and 1.5mm gap. (page: 4, paragraph 1)

Also, I would recommend to describe in details WMH volume analysis with MRIcro software in 2.4 Neuroimaging.

Thank you for the recommendation. We added details in the description of WMH volume analyses with MRIcro:

WMH volume (WMHv) analysis was performed on axial FLAIR sequences using a MRIcro software, according to previously validated methods[23]. FLAIR and the DWI sequences were aligned to exclude acute and chronic infarcts. Using operator-mediated quality assurances, overlapping regions of interest (ROIs) corresponding to WMH produced the final maps for WMH volume calculation. To correct WMH volume for head size, we used the sagittal midline cross-sectional intracranial area (ICA) as surrogate measure of the intracranial volume. WMH was normalized multiplying the measured WMH by the ratio of the individual ICA to mean ICA of the whole cohort. Supratentorial WMH volume was given as log-transformed. All supratentorial white matter and deep grey -matter lesions were included, with the exception of WMH corresponding to infarcts. To further avoid confusion, we measured only WMH from the hemisphere unaffected by stroke and doubled this value to calculate total WMHv. All readers have previously shown high interrater agreement for determination of WMHv (free marginal ҡ>0. 9 0). All MRI measurements were performed by readers blinded to clinical data. (page: 4, paragraph 2)
